# On the Efficiency of Transformers: The Effect of Attention Rank

## Abstract

Transformers have showcased superior performance across a variety of real-world scenarios, marking the advent of a period dominated by large language models. However, with the escalating complexity of these models and the continuous enlargement of training datasets, efficiency-related challenges have become more pronounced. In this study, we investigate the influence of the rank of attention matrices on the training and performance of these models. We first gain insight by benchmark tasks such as BERT and GPT-2. Our findings underscore that (i) the mean rank of attention matrices is stable throughout the training, and the initial rank is a dependable indicator of the final rank; (ii) a distinct positive relationship exists between the attention rank and the effectiveness of the model, where elevated ranks correlate with diminished loss and expedited convergence. These insights reveal a relationship between initial attention matrix rank and performance. We proceed to investigate the impact of hyperparameters on the initial rank. The study unveils that modifying the softmax temperature or the head dimension can amplify the ranks, with the former exerting a more significant effect. Notably, we theoretically identify the characterization in the attention matrix rank at low temperatures, and we demonstrate the existence of an upper bound of attention matrix rank with respect to the head dimension. These observations are validated through trials on a high-rank target, underscoring instances where traditional setups fall short.

## 1 Introduction

In recent years, Transformer-based neural network models have reshaped the landscape of machine learning, demonstrating unparalleled success across a myriad of applications including natural language processing (NLP) Vaswani et al. (2017); Devlin et al. (2019); Raffel et al. (2020); Radford et al. (2018); Rae et al. (2021); Dehghani et al. (2023); Touvron et al. (2023); Liu et al. (2019); Hao et al. (2020); Liu et al. (2021); Yuan et al. (2022), computer vision (CV) Chen et al. (2021b); Wang et al. (2022); Liang et al. (2021); Lu et al. (2022); Zhu et al. (2021); Wang et al. (2021), audios Sung et al. (2022); Tsimpoukelli et al. (2021); Li et al. (2022), interdisciplinary sciences Jumper et al. (2021), and so on. Their core architecture module, anchored by the so-called attention mechanism, has been proved to be a cornerstone particularly in capturing linguistic relationships with intricacies and nuances, thereby driving the current NLP renaissance and leading to the new era of large foundation models represented by ChatGPT and GPT-4. However, in the meantime, with this rise in prominence comes pressing challenges. As model architectures burgeon in complexity and training data swells in volume, the looming issue of *efficiency* becomes highly inescapable Shen et al. (2023).

In this study, we carry out a thorough investigation on the *rank* properties of the Transformer model. Mathematically, the central attention mechanism is designed to weigh the significance and correlations of input tokens via, e.g., inner products between trainable transformations on inputs, which is often formulated as attention matrices. As a fundamental concept in algebra, the matrix rank is supposed to impact the capacity (expressive ability) and learning performance of the attention mechanism and hence Transformer models. However, although there are findings on the low-rank bottleneck Kanai et al. (2018); Bhojanapalli et al. (2020); Dong et al. (2021); Lin et al. (2022), and several Transformer-based variants to reduce the computational and memory bottlenecks of modeling long sequences from the perspective of attention rank Chen et al. (2021a); Wang et al. (2020); Hu

et al. (2022); Guo et al. (2019); Lin et al. (2022), the effect of attention ranks has been overshadowed by other model intricacies to a large extent.

In this regard, one may pose the following fundamental questions:

1. Intuitively, both the attention rank and model performance vary during the training process. How do they affect and evolve with each other?

2. What is the influence of hyperparameters configuration in attention matrices on the attention rank, and hence the model performance and parameter efficiency?

3. Can these findings and insights provide tutorial guidance in practical applications?

We develop principled results on the first and second questions via systematic experiments and rigorous mathematical analysis and make further steps on the third question by numerical verifications under toy but representative scenarios. The primary contributions can be summarized as follows:

1. **Stability of attention rank**: We present empirical evidence on real-world NLP benchmarks showcasing that the rank of attention matrices remains almost constant during the training process. This makes the initial attention rank an applicable and convenient measure of the attention rank along with training.

2. **Connections between attention rank and performance**: Our findings illuminate a direct and positive correlation between the attention rank and model performance. Specifically, a higher rank usually leads to expedited convergence and decreased loss, especially when learning high-rank targets. Combined with Point 1, this emphasizes the importance of the initial rank. Consequently, these insights not only streamline the model design process as well as the hyperparameter selection but also contribute to conserving valuable computational resources.

3. **Effect of softmax temperature and head dimension on attention rank**: We provide a fine-grained analysis of factors affecting the (initial) attention rank. Notably, it is shown that while both the softmax temperature and head dimension play a role, the impact of temperature is much more pronounced.

4. **Theoretical demonstration**: Under the setting of reduced temperatures, we perform rigorous mathematical analysis on the rank of attention matrices. The results (i) establish an upper bound on the (initial) attention rank, suggesting the existence of low-rank limits; (ii) imply a model reduction effect corresponding to parameter efficiency. That is, it is sufficient for the attention rank to reach saturation given a relatively small head dimension.

5. **Validations**: We numerically verify the results under a controlled but representative setting, where challenges that may be encountered in real-world tasks are mainly emphasized via target ranks. This validation underscores the applicability and robustness of our findings and insights.

The rest of this paper is organized as follows. In Section 2, we discuss the related work centering around the attention rank. Section 3 provides pivotal findings in real-world applications (BERT and GPT-2 on benchmark datasets). Section 4 includes the fine-grained mathematical analysis on the attention rank. In Section 5, we perform numerical verifications to validate our results and insights. All the details of proofs can be found in the appendix.

**Notations.** Throughout this paper, we use normal letters to denote scalars, particularly the letters $n, d, d_h, i, j, k$ to represent positive integers. Boldface lower-case/capital letters are reserved for vectors/matrices. Let $\|\mathbf{x}\|_p := \left(\sum_{i=1}^{n} x_i^p\right)^{1/p}$ be the $\ell^p$-norm for any $\mathbf{x} \in \mathbb{R}^n$ and $p \in [1, \infty]$. Denote the standard basis of $\mathbb{R}^n$ by $\{\mathbf{e}_i\}_{i=1}^{n}$, where $\mathbf{e}_i$ is the vector of all zeros except that the $i$-th position is 1. Let $\mathbf{0}_n \in \mathbb{R}^n$ be the vector of all zeros. Let $[n] := \{1, 2, \ldots, n\}$, $n \in \mathbb{N}_+$. For a probability space $(\Omega, \mathcal{F}, \mathbb{P})$, denote the probability of a measurable event $E \in \mathcal{F}$ by $\mathbb{P}(E)$. Let $\mathcal{N}(\boldsymbol{\mu}, \boldsymbol{\Sigma})$ be the multivariate normal distribution defined on $\mathbb{R}^n$, where $\boldsymbol{\mu} \in \mathbb{R}^n$ is the expectation and $\boldsymbol{\Sigma} \in \mathbb{R}^{n \times n}$ is the covariance. We use the big Omega notation $f(n) = \Omega(g(n))$ to represent that $f$ is bounded below by $g$ asymptotically, i.e., there exists $c > 0, n_0 \in \mathbb{N}_+$ such that $f(n) \geq cg(n)$ for any $n \geq n_0$.

## 2 RELATED WORK

The exploration of the rank of the Transformer attention matrix has been a focus in previous research (Kanai et al. (2018); Bhojanapalli et al. (2020); Dong et al. (2021); Lin et al. (2022)). Bhojanapalli et al. (2020) unveiled a restriction associated with the low-rank bottleneck in attention heads, attributed to the proportional relationship between the number of heads and the size of each head in prevailing architectures. Dong et al. (2021) introduced an innovative perspective of interpreting self-attention networks. Their study elucidated that the networks' output is an amalgamation of lesser components, or pathways. In the absence of skip connections and multi-layer perceptrons (MLPs), they established that the output gravitates towards a rank-1 matrix at a doubly exponential rate.

On the other hand, a suite of Transformer-based adaptations (Chen et al. (2021a); Wang et al. (2020); Hu et al. (2022); Guo et al. (2019); Lin et al. (2022)) has emerged to mitigate the inherent bottlenecks, notably computational and memory constraints. For instance, Wang et al. (2020) ascertained that the self-attention mechanism's complexity is reducible, attributing this to its low-rank matrix approximation. The innovative self-attention mechanism they introduced marked a reduction in complexity. Meanwhile, Guo et al. (2019) incorporated low-rank constraints, a modification that manifested improvements in specific tasks. In a parallel vein, Chen et al. (2021a) noted the prowess of sparse and low-rank approximations in distinct scenarios. Their efficacy was found to be contingent on the softmax temperature in attention, with a combined sparse and low-rank approach superseding individual performances.

In the context of our research, a meticulous analysis of attention matrices' rank and its bearing on model efficiency and performance is conducted. We establish that the mean rank remains consistent throughout the training, positioning the initial rank as an accurate predictor of the end rank. Furthermore, a clear linkage is discerned between increased attention ranks and a reduction in loss and accelerated convergence, especially for high-rank targets. Delving into the impact of different configurations on the initial rank, we observe that both the softmax temperature and head dimension ($d_h$) adjustments lead to augmented ranks. The softmax temperature adjustments are particularly prominent. At lower temperatures, a unique attention matrix rank pattern emerges. Our theoretical insights, corroborated by experimental assessments on an optimized model, accentuate the limitations inherent in traditional configurations, underscoring the pivotal role of these parameters.

## 3 ATTENTION RANK IN BERT AND GPT-2

In the realm of Natural Language Processing (NLP), transformer-based models have risen to prominence, with the self-attention mechanism being instrumental in their ascendancy by offering enhanced handling of sequential data. We first delve into an analytical comparison of two renowned models, BERT and GPT-2, with a particular focus on attention matrix rank. The rank of a matrix, a concept central to linear algebra, serves as a critical element in our analysis, offering insights into the amount of distinct information encapsulated within the matrix. In the context of transformer models, understanding the rank of the attention matrix is crucial, as it potentially correlates with the model's performance.

### 3.1 FORMULATIONS

To delve deeper into the intricacies of this mechanism, we commence with its mathematical formulations. A transformer consists of several interconnected transformer blocks. For each head $h \in \{1, 2, \cdots, H\}$ in each block, we have

$$\mathbf{V}^h = \mathbf{X}\mathbf{W}_v^h, \quad \mathbf{K}^h = \mathbf{X}\mathbf{W}_k^h, \quad \mathbf{Q}^h = \mathbf{X}\mathbf{W}_q^h, \tag{1}$$

where $\mathbf{W}_v^h, \mathbf{W}_k^h, \mathbf{W}_q^h \in \mathbb{R}^{d \times d_h}$, and $d_h$ is the head dimension with $d_h = d/H$. We denote the input by $\mathbf{X} \in \mathbb{R}^{n \times d}$. The attention (score) matrix is formulated as

$$\mathbf{Attn}^h = \text{softmax}\left(\frac{\mathbf{Q}^h(\mathbf{K}^h)^\top}{T}\right), \tag{2}$$

where $T > 0$ is the temperature, and is typically assigned the value $\sqrt{d_h}$ in most applications. The subsequent output is

$$\mathbf{O}_{\text{attn}} = \mathbf{LN}(\text{Concatenate}_{[H]}(\mathbf{Attn}^h \mathbf{V}^h)\mathbf{W}_o + \mathbf{X}) \tag{3}$$

with $\mathbf{W}_o \in \mathbb{R}^{d \times d}$ and $\mathbf{LN}$ as the layer normalization. This yields the final output

$$\mathbf{O}_{\text{output}} = \mathbf{FFN}(\mathbf{O}_{\text{attn}}), \tag{4}$$

where $\mathbf{FFN}$ is the feedforward neural network.

From the formulation, we find that the head dimension $d_h$ and the temperature $T$ largely dictate the rank of attention matrices. We will show that these hyperparameters play a vital role in determining both the representational prowess and efficiency of the attention mechanism and transformer model later.

### 3.2 EXPERIMENTS

**Task.** Our experiments were initiated with a focus on the key parameters $d_h$ and $T$, owing to their critical role in the rank of the attention matrix. We employed BERT and GPT-2 models for this purpose. The core aim was to investigate how alterations to $d_h$ and $T$ would influence the rank in training dynamics and, subsequently, on the overall performance of these BERT and GPT-2 models. The IMDB and Wiki datasets served as the data for our training processes.

**Model and hyper-parameters.** We utilized transformer models composed of 6 layers, with an embedding size of 256. Training batches were made up of 8 samples and employed a learning rate of $5 \times 10^{-5}$. In order to evaluate the rank, we chose four random samples each from the training and testing datasets and computed the average attention rank across all transformer blocks and heads. We also monitored the variance between head and data samples to gain insights into the attention mechanism's stability and consistency.[1]

Visual representations of our experimental outcomes are delineated in Figures 1 - 4, offering an illustrative analysis that aids in comprehensively understanding the effects of variations in $d_h$ and $T$ on the rank in the training dynamics and final performance.

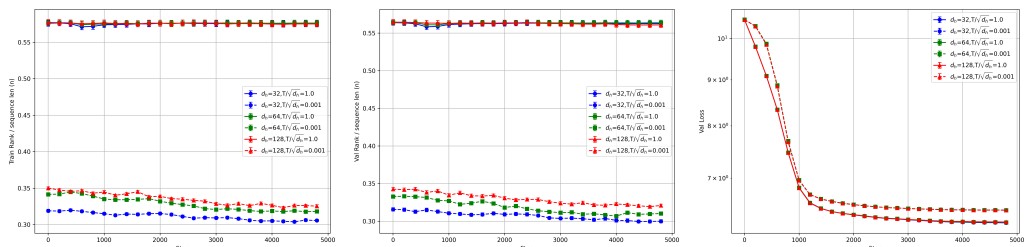

Figure 1: BERT on IMDB. From left to right: training rank, validation rank, and validation loss.

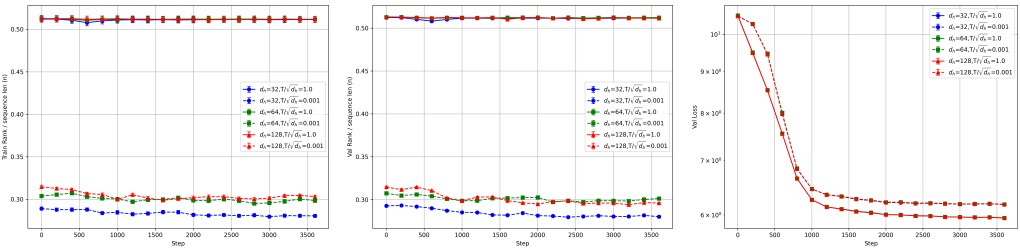

Figure 2: BERT on Wiki. From left to right: training rank, validation rank, and validation loss.

---

[1] It should be noted that the training of GPT-2 necessitated the masking of attention matrices to preclude the model from accessing future data. However, this mask was not applied during the rank evaluation.

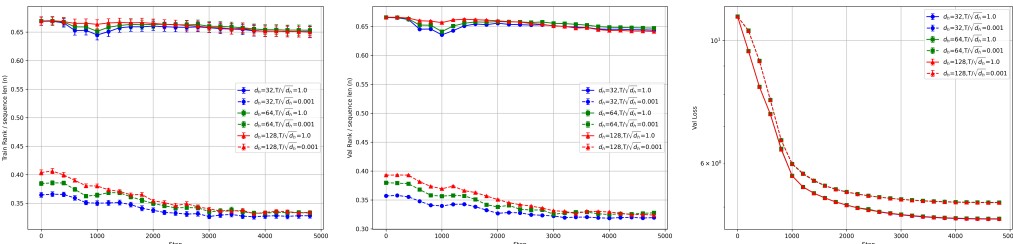

Figure 3: GPT-2 on IMDB. From left to right: training rank, validation rank, and validation loss.

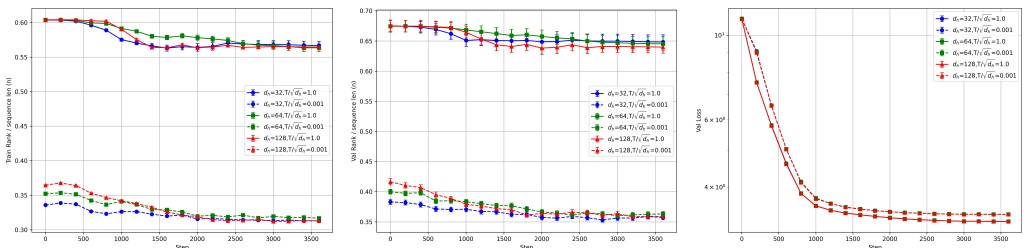

Figure 4: GPT-2 on Wiki. From left to right: training rank, validation rank, and validation loss.

## 3.3 DISCUSSIONS

The graphical representations and data derived from our experiments yield several key insights into the behavior of the attention matrix rank. We distill our primary observations as follows:

1. **Rank Stability Throughout Training**: One salient observation is the stable nature of the attention matrix rank during the training process. Across diverse experiments, the rank exhibits minimal fluctuations, underscoring that the initial rank profoundly impacts subsequent training phases. This stability in both the training and validation phases accentuates the critical role of model initialization.

2. **Role of Temperature** ($T$): Alterations in $T$ markedly impact the initial rank of the attention matrix. Across all values of $d_h$, an increase in $T$ correlates with a higher rank. A distinct divergence is observed between the rank curves corresponding to $T = 0.001$ and $T = 1$, highlighting the pivotal role of temperature in shaping the matrix's initial structure and, by extension, its representational capabilities. This underscores the imperative of judiciously choosing the $T$ value.

3. **Effect of Head Dimension** ($d_h$): In contrast to $T$, variations in $d_h$ exert a less pronounced impact on the attention matrix rank. A slight elevation in rank is observed when $d_h$ ascends from 32 to 64. However, this elevation becomes marginal when $d_h$ escalates from 64 to 128, despite the absolute increment in $d_h$ being larger, especially when $T$ is larger. This implies a diminishing return in rank enhancement as $d_h$ increases and underscores that the influence of $d_h$ is comparatively subdued relative to $T$ variations.

4. **Association with Model Performance**: A higher attention matrix rank correlates with enhanced model efficacy, as manifested by reduced validation loss. Models conditioned with $T = 1$ consistently eclipse those conditioned with $T = 0.001$, registering lower validation losses. Interestingly, the disparity in attention matrix ranks for different $d_h$ values (32, 64, 128) is negligible, and the performance of the corresponding models is almost analogous.

These observations are invariant across multiple experiments, lending credence to the universality of these insights.

## 4 FINE-GRAINED THEORETICAL ANALYSIS

As is shown in Figures 1 - 4 and discussed in Section 3.3, the attention rank largely determines the overall model performance, with a higher initial rank leading to reduced loss and faster convergence. In addition, compared to the head dimension, the softmax temperature has a much more pronounced impact on the (initial) attention rank. This section provides a fine-grained analysis of the low-temperature case associated with the "hardmax" activation,[2] illustrating the existence of the low-rank barrier and model reduction effect.

**Formulations.** Let $\mathbf{X} := [\mathbf{x}_1, \mathbf{x}_2, \ldots, \mathbf{x}_n]^\top \in \mathbb{R}^{n \times d}$ be the input data, where $n$ denotes the sequence length and $d$ is the input dimension. Let $(\mathbf{K}, \mathbf{Q}) = (\mathbf{X}\mathbf{W}_k, \mathbf{X}\mathbf{W}_q)$ be the key-query pair with trainable parameters $\boldsymbol{\theta} := (\mathbf{W}_k, \mathbf{W}_q) \in \mathbb{R}^{d \times d_h} \times \mathbb{R}^{d \times d_h}$ ($d_h$ is the head dimension), i.e., $\mathbf{K} := [\mathbf{k}_1, \mathbf{k}_2, \ldots, \mathbf{k}_n]^\top \in \mathbb{R}^{n \times d_h}$, $\mathbf{Q} := [\mathbf{q}_1, \mathbf{q}_2, \ldots, \mathbf{q}_n]^\top \in \mathbb{R}^{n \times d_h}$ with $\mathbf{k}_i^\top = \mathbf{x}_i^\top \mathbf{W}_k$, $\mathbf{q}_i^\top = \mathbf{x}_i^\top \mathbf{W}_q$, $i = 1, 2, \ldots, n$. The basic form of the self-attention (score) matrix is defined as

$$\mathbf{Attn}(\mathbf{X}; \boldsymbol{\theta}) := \mathrm{softmax}\left(\mathbf{Q}\mathbf{K}^\top / T\right) = \mathrm{softmax}\left(\mathbf{X}\mathbf{W}_q \mathbf{W}_k^\top \mathbf{X}^\top / T\right), \tag{5}$$

where $T := T(n, d, d_h) > 0$ is the temperature. By convention, for any $\mathbf{A} = [a_{ij}] \in \mathbb{R}^{n \times n}$, $\mathbf{e}_i^\top \mathrm{softmax}(\mathbf{A})\mathbf{e}_j := \frac{\exp(a_{ij})}{\sum_{j=1}^n \exp(a_{ij})}$ with $\{\mathbf{e}_i\}_{i=1}^n$ as the standard basis of $\mathbb{R}^n$.

For the low-temperature case ($0 < T \ll 1$), (5) is approximately

$$\mathrm{hardmax}\left(\mathbf{X}\mathbf{W}_q \mathbf{W}_k^\top \mathbf{X}^\top\right). \tag{6}$$

See Lemma 1 in the appendix for further details. Here, the maximum is taken in a row-wise sense: for a matrix $\mathbf{A} = [a_{ij}] \in \mathbb{R}^{n \times n}$, $\mathbf{e}_i^\top \mathrm{hardmax}(\mathbf{A}) := \mathbf{e}_{k_i}$ with $k_i := \arg\max_{j \in [n]} a_{ij}$.

We have the following estimate on the rank of (6).

**Theorem 1.** *Let the parameters $\mathbf{W}_q, \mathbf{W}_k$ be Gaussian random matrices, i.e., the entries of $\mathbf{W}_q, \mathbf{W}_k$ are independent $\mathcal{N}(0, 1)$ random variables. Assume that the input data $\mathbf{X}$ satisfies $\mathbf{X}\mathbf{X}^\top = \mathbf{I}_n$.[3] Then for any $n \in \mathbb{N}_+$ appropriately large, we have*

$$\mathbb{E}_{\mathbf{W}_q}\left[\mathrm{rank}\left(\mathrm{hardmax}\left(\mathbf{X}\mathbf{W}_q \mathbf{W}_k^\top \mathbf{X}^\top\right)\right)\right] \le (1 - \exp(-1))n + 1 \approx 0.63n. \tag{7}$$

**Proof sketch.** The theorem is proved via the following procedure:

1. The orthonormal input yields the independence across different rows of $\mathbf{X}\mathbf{W}_q \mathbf{W}_k^\top \mathbf{X}^\top$ (Lemma 4 in the appendix), further implying that these rows are i.i.d. as $\mathcal{N}(\mathbf{0}_n, \mathbf{Q}\mathbf{K}^\top)$ for any fixed (Gaussian random) $\mathbf{W}_k$.

2. The hardmax calculation within respective rows is reduced to an elementary birthday problem (Lemma 3 in the appendix), which gives the estimate on the number of columns with all zeros;

3. The estimate is further analyzed via the infinitesimal order estimation (Lemma 2 in the appendix) and the AM-GM inequality (suggesting that "=" holds if and only if all probabilities are equal).

**Remark 1.** *The assumption on input data seems strong at first glance. However, this assumption is approximately reasonable in applications where different $\boldsymbol{x}_i$'s (corresponding to different tokens, for example) are often embedded with independent (and isotropic) Gaussian vectors. According to Vershynin (2018) (Lemma 3.2.4 and Remark 3.2.5), $\boldsymbol{x}_i$'s tend to be almost orthogonal in high dimensions ($d \gg 1$) after proper scaling (e.g., normalization).*

**Remark 2.** *Notice that the $\mathrm{hardmax}(\cdot)$ operator is scaling-invariant w.r.t. positive constants, i.e., $\mathrm{hardmax}(c\boldsymbol{A}) = \mathrm{hardmax}(\boldsymbol{A})$ for any $c > 0$. Theorem 1 also holds when the input sequences are not normalized.*

---

[2]The hardmax activation is occasionally used in applications for computational efficiency. See CV examples in Elsayed et al. (2019); Papadopoulos et al. (2021) for more details.

[3]That is, the input sequence is orthonormal across time steps.

**The model reduction effect.** In fact, the above rank (LHS of (7)) reaches saturation when increasing the head dimension $d_h$, provided an appropriate scaling (e.g., $1/\sqrt{d_h}$). Informally, recall that the rows of $\boldsymbol{X}\boldsymbol{W}_q\boldsymbol{W}_k^\top\boldsymbol{X}^\top$ are independent and identically distributed as $\mathcal{N}(\boldsymbol{0}_n, \boldsymbol{K}\boldsymbol{K}^\top)$, according to the Johnson–Lindenstrauss lemma (Johnson & Lindenstrauss (1984)). We have

$$\boldsymbol{e}_i^\top \boldsymbol{K}\boldsymbol{K}^\top \boldsymbol{e}_j = \boldsymbol{k}_i^\top \boldsymbol{k}_j = \boldsymbol{x}_i^\top \boldsymbol{W}_k \boldsymbol{W}_k^\top \boldsymbol{x}_j$$
$$\approx \boldsymbol{x}_i^\top \boldsymbol{x}_j \Rightarrow \boldsymbol{K}\boldsymbol{K}^\top \approx \boldsymbol{X}\boldsymbol{X}^\top, \quad \text{when } d_h = \Omega(\log n). \tag{8}$$

That is, further increasing the head dimension after $d_h = \Omega(\log n)$ has limited effect on the rows' distribution (always approximately $\mathcal{N}(\boldsymbol{0}_n, \boldsymbol{X}\boldsymbol{X}^\top)$ only depending on $n, d$), and hence on the rank of hardmax attention. This is the model reduction effect: selecting the critical configuration $d_h = \Omega(\log n)$ achieves optimal efficiency, since further increasing parameters leads to diminishing marginal utility.

## 5 NUMERICAL VERIFICATIONS

Building on the insights from our previous experiments and theoretical results, we established that the rank of the attention matrix is crucial in determining the model's overall performance. Notably, the initial rank predominantly governs the rank throughout the training process, underscoring the significance of model initialization. Now, we aspire to validate our theoretical analysis results and verify the impact of attention rank on model performance in a more controlled data environment.

### 5.1 INITIAL ATTENTION RANK

**Task.** We first focus on the attention matrix at initialization. Recall that

$$\textbf{Attn} = \text{softmax}\left(\frac{\mathbf{X}\mathbf{W}_q(\mathbf{X}\mathbf{W}_k)^\top}{T}\right), \tag{9}$$

where $\mathbf{X} \in \mathbb{R}^{n \times d}$ and the elements of $\mathbf{X}$, $\mathbf{W}_q$, and $\mathbf{W}_k$ are drawn from a $\mathcal{N}(0, 1)$ distribution. Our aim is to explore and understand the behavior of the attention matrix's rank at different values of the temperature parameter $T$ and the dimensionality $d_h$.

**Model and Hyperparameters.** For our assessment, we set $n = 100$ and $d = 256$. We test $d_h$ at values $\{8, 16, 32, 64, 128\}$ and $T$ across a logarithmic scale from $10^{-4}$ to $10^3$. Employing singular value decomposition, we ascertain the matrix rank, treating near-zero singular values as zero with a threshold of $10^{-8}$.

In order to bolster the reliability of our findings, we calculate the matrix rank for each $d_h$ and $T$ combination over three trials, and compute the mean and standard deviation. This systematic approach ensures a comprehensive analysis, providing insights that are both robust and replicable. Results are presented in Figure 5.

Figure 5 reveals salient trends concerning the matrix rank, temperature parameter $T$, and diverse $d_h$ values. Significantly, the matrix rank demonstrates acute sensitivity to temperature $T$. For all $d_h$ values, an increase in $T$ leads to a marked rise in matrix rank, eventually achieving full rank. This pattern suggests that matrices at higher temperatures generally maintain a superior effective rank, emphasizing the pivotal role of $T$ in the attention mechanism. Elevated temperatures result in a wider attention distribution, potentially amplifying the attention matrix's rank. For temperatures below 10, the matrix primarily manifests a low-rank property, implying limited expressiveness. Intriguingly, for all $d_h$ values, minimal rank variance is observed in the attention matrix around $T = 10$ and at particularly low $T$ values (around $10^{-4}$). At these exceedingly low $T$ values, the behavior resembles hardmax. Hence, from a rank perspective, conditions with $T < 10$ mirror the hardmax scenario. Beyond $T = 10$, the rank rapidly surges with increasing $T$, ultimately reaching full rank.

In contrast, although $d_h$ determines the maximum achievable rank for the $\mathbf{W}_q$ and $\mathbf{W}_k$ matrices, and it indirectly regulates the overall rank of the attention matrix, the impact of $d_h$ on rank is far less significant than that of $T$. Observing settings with very low temperatures, we find that (i) there exists an upper bound for the attention rank, which is *consistent* with our theoretical estimate

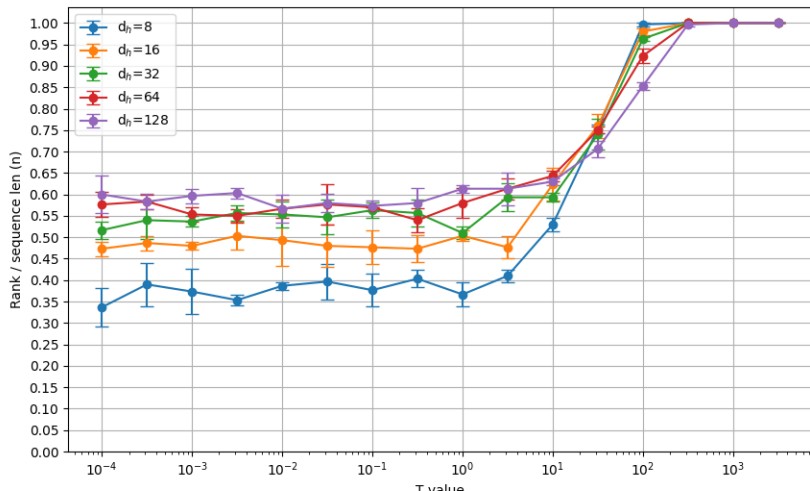

Figure 5: The variation of (initial) attention rank for different $d_h, T$ values, where error bars represent the standard deviation.

(approximately $0.63n$); and (ii) when the head dimension $d_h$ is not too large (much smaller than $n$), the attention rank has reached saturation, *aligning* with our theoretical predictions as well.

The error bars, symbolizing the standard deviation, convey inherent variations. Yet, these variations are eclipsed by the predominant patterns. In most instances, the standard deviation is trivial, signifying a consistent trend. This consistency emphasizes the dependability of the identified patterns and trends. Our findings elucidate the intricate interplay between temperature $T$ and $d_h$ in determining the attention matrix's effective rank. Such insights could offer invaluable guidance for model architecture design, guiding choices about the optimal temperature and number of attention heads to enhance performance for particular tasks or datasets.

**Remark 3.** *It is noteworthy that while the trend of rank variation with $T$ concurs with earlier observations in GPT-2 and BERT, where the rank ascends with increasing $T$, in our simplified setting, the maximum rank can achieve full rank. In contrast, in GPT-2 and BERT experiments, full rank is never attained. This discrepancy arises from the inherently low-rank characteristic of real text data, marked by frequent appearances of common words like "a" and "the." Additionally, the $T$ scale observed in our experiments differs from that in GPT-2 and BERT, as these models' actual training employed Kaiming initialization instead of the standard $\mathcal{N}(0, 1)$ initialization.*

## 5.2 EXPERIMENTS ON HIGH-RANK DATA

**Task.** In our pursuit to delve deeper into the effects of the attention matrix's rank on the model's expressive capability, a simplified experiment was conducted in a more controlled data environment, emphasizing the influence of its initial rank on the model's performance on high-rank signals. We constructed a high-rank dataset that closely mirrors our conditions of interest. The sequence $\mathbf{X}$ is consistently composed of 100 characters. Characters were randomly and uniformly selected from a predefined set to ensure a diverse array of sequences. The sequence $\mathbf{Y}$ is formulated using the equation $\mathbf{Y} = \mathbf{XP}$, where $\mathbf{P}$ is a $100 \times 100$ matrix, and each row in $\mathbf{P}$ contains a single '1', with all other elements set to '0'. We fixed the rank of $\mathbf{P}$ at 80 to simulate a high-rank target, yielding a dataset comprising 5000 data pairs $(\mathbf{X}, \mathbf{Y})$.

**Model and Hyperparameters.** Our model begins with an embedding layer that transposes the input into a dense vector space, followed by a transformer block that encapsulates key elements like multi-head self-attention, position-wise feed-forward networks, skip connections, and layer normalization. We set $n = 100$ and $d = 256$. We test $d_h$ at values $\{32, 64, 128\}$ and $T$ at values

Table 1: Performance across different hyperparameter configurations and varied attention ranks.

| $T$ | $d_h$ | Final Loss | Initial Rank | Final Rank | Accuracy |
|---|---|---|---|---|---|
| 1 | 32 | 0.52 | $54.29 \pm 0.44$ | $52.25 \pm 0.44$ | 0.94 |
| 1 | 64 | 0.72 | $57.70 \pm 0.40$ | $56.88 \pm 0.40$ | 0.93 |
| 1 | 128 | 0.91 | $59.90 \pm 0.61$ | $59.05 \pm 0.61$ | 0.92 |
| 100 | 32 | 0.00 | $95.62 \pm 0.06$ | $95.38 \pm 0.06$ | 1.00 |
| 100 | 64 | 0.00 | $91.40 \pm 0.29$ | $90.17 \pm 0.29$ | 1.00 |
| 100 | 128 | 0.06 | $85.35 \pm 2.05$ | $78.25 \pm 2.05$ | 0.99 |
| 1000 | 32 | 0.00 | $100.00 \pm 0.00$ | $100.00 \pm 0.00$ | 1.00 |
| 1000 | 64 | 0.00 | $100.00 \pm 0.00$ | $100.00 \pm 0.00$ | 1.00 |
| 1000 | 128 | 0.00 | $100.00 \pm 0.00$ | $100.00 \pm 0.00$ | 1.00 |

$\{1, 100, 1000\}$. According to Figure 5, different values of $T$ result in different initial attention matrix ranks. Specifically, with $T = 1$ (the usual setting), the initial attention matrix rank is less than the data's rank (80), while with $T = 100$ and $T = 1000$, the initial attention matrix rank exceeds the data's rank (80).

The forward pass of the model culminates in a linear layer that delivers the final predictions, ensuring that the output is aligned with the expected results. Optimization is facilitated by the Adam optimizer, with a learning rate of 0.003. We gauge the model's evolution through the cross-entropy loss, a conventional metric for classification tasks. The experiment is characterized by an embedding size of 256, a total of 50 training epochs, and a batch size of 200. The findings are illustrated in Table 1.

## 5.3 DISCUSSIONS

Our refined model unveiled insights that affirm our experimental discoveries. First, the influence of $T$ and $d_h$ on the final rank is consistent with their impact on the initial rank, as observed in Figure 5. This consistency further illustrates the stable rank of the attention matrix throughout training. For example, at $T = 1$ and $d_h = 32$, the rank slightly adjusts from $54.29 \pm 0.44$ to $52.25 \pm 0.44$. A similar steadiness is reflected in other settings, consistent with our experimental observations on GPT-2 and BERT.

Further, we find a discernible correlation between the attention matrix rank and the model's overall performance. Setups where the rank exceeds the target rank (80) witness a significant enhancement in model efficacy. Specifically, setups with $T = 100$ and $T = 1000$ achieve an accuracy approaching 1.00, markedly superior to those at $T = 1$. Interestingly, variations in $d_h$ yield almost imperceptible performance differences, highlighting the attention matrix rank's potential as a powerful early indicator for assessing model efficacy.

## 6 CONCLUSION

This study examined the stability of attention rank and its association with model performance and efficiency. Our empirical results are bolstered by comprehensive mathematical analysis and numerical validation. Initially, we emphasized the consistent behavior of attention rank throughout training. Additionally, we found that the initial attention rank plays a pivotal role in determining the ultimate performance of the model.

Moreover, we identified the substantial impact of softmax temperature and head dimension on attention rank, with temperature having a more dominant effect. These observations are crucial for enhancing model performance and efficiency. Future research will expand on these insights, delving further into the intricacies of attention mechanisms and revealing potential avenues for advanced applications.

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

## A  APPENDIX

We begin with a lemma characterizing the gap between the softmax function and its "hard" version.

**Lemma 1.** *Let $\boldsymbol{a} = [a_1, a_2, \cdots, a_n]^\top \in \mathbb{R}^n$ with $i^* := \arg\max_{i \in [n]} a_i$ and $i'^* := \arg\max_{i \in [n], i \neq i^*} a_i$, and* $\mathrm{hardmax}(\boldsymbol{a}) := \boldsymbol{e}_{i^*}$. *Assume that $\delta := a_{i^*} - a_{i'^*} > 0$ (i.e., the maximum is unique). Then for any $T > 0$, we have*

$$\Delta_{n,\delta}(T) := \|\mathrm{softmax}(\boldsymbol{a}/T) - \mathrm{hardmax}(\boldsymbol{a})\|_1 \leq 2(n-1)\exp(-\delta/T). \tag{10}$$

*That is, $\Delta_{n,\delta}(T)$ converges to 0 exponentially fast as $T \to 0^+$.*

*Proof.* It is straightforward to have

$$
\begin{aligned}
\Delta_{n,\delta}(T) &= \sum_{i \in [n], i \neq i^*} \frac{\exp(a_i/T)}{\sum_{j=1}^n \exp(a_j/T)} + 1 - \frac{\exp(a_{i^*}/T)}{\sum_{j=1}^n \exp(a_j/T)} \\
&= 2\frac{\sum_{i \in [n], i \neq i^*} \exp(a_i/T)}{\sum_{i \in [n], i \neq i^*} \exp(a_i/T) + \exp(a_{i^*}/T)} \\
&\leq 2 \sum_{i \in [n], i \neq i^*} \exp((a_i - a_{i^*})/T) \\
&\leq 2(n-1)\exp((a_{i'^*} - a_{i^*})/T) \\
&= 2(n-1)\exp(-\delta/T).
\end{aligned}
$$

This gives $\lim_{T \to 0^+} \Delta_{n,\delta}(T) = 0$, and the rate is exponentially fast. The proof is completed. □

According to Lemma 1, for the low-temperature case where $T \ll 1$, one can approximately consider the hard version

$$\text{hardmax}\left(\boldsymbol{X}\boldsymbol{W}_q\boldsymbol{W}_k^\top\boldsymbol{X}^\top\right), \tag{11}$$

where the maximum is taken row-wisely. That is, for any $\boldsymbol{A} = [a_{ij}] \in \mathbb{R}^{n \times n}$, $\boldsymbol{e}_i^\top \text{hardmax}(\boldsymbol{A}) := \boldsymbol{e}_{k_i}$ with $k_i := \arg\max_{j \in [n]} a_{ij}$. Note that the $\text{hardmax}(\cdot)$ operator is (positively) scaling-invariant, i.e. $\text{hardmax}(c\boldsymbol{A}) = \text{hardmax}(\boldsymbol{A})$ for any $c > 0$.

Before we prove the low-rank property of (11), the following elementary lemmas are useful.

**Lemma 2.** *For any $c > 0$, we have*

$$\lim_{y \to 0} \frac{(1 - cy)^{\frac{1}{y}} - \exp(-c)}{y} = -\frac{c^2}{2}\exp(-c). \tag{12}$$

*Proof.* By L'Hôpital's rule, we have

$$
\begin{aligned}
\lim_{y \to 0} \frac{(1 - cy)^{\frac{1}{y}} - \exp(-c)}{y} &= \lim_{y \to 0} (1 - cy)^{\frac{1}{y}} \cdot \frac{-\log(1 - cy) - \frac{cy}{1-cy}}{y^2} \\
&= -\exp(-c) \cdot \lim_{y \to 0} \frac{\log(1 - cy) + \frac{cy}{1-cy}}{y^2} \\
&= -c\exp(-c) \cdot \lim_{y \to 0} \frac{\frac{-1}{1-cy} + \frac{1}{(1-cy)^2}}{2y} \\
&= -c\exp(-c) \cdot \lim_{y \to 0} \frac{\frac{-c}{(1-cy)^2} + \frac{2c}{(1-cy)^3}}{2} \\
&= -\frac{c^2}{2}\exp(-c), \tag{13}
\end{aligned}
$$

which gives the desired result. $\square$

**Lemma 3.** *For a random matrix $\boldsymbol{A} = [a_{ij}] \in \mathbb{R}^{n \times n}$ with independent rows, let $p_{ij} := \mathbb{P}(\{a_{ij} = \max_{j' \in [n]} a_{ij'}\})$. Then the expectation number of columns with all zeros in $\text{hardmax}(\boldsymbol{A})$ is*

$$\sum_{j=1}^{n} \prod_{i=1}^{n} (1 - p_{ij}). \tag{14}$$

*Proof.* Define the random variable

$$X_j = \begin{cases} 1, & \text{hardmax}(\boldsymbol{A})\boldsymbol{e}_j = \boldsymbol{0}_n, \\ 0, & \text{hardmax}(\boldsymbol{A})\boldsymbol{e}_j \neq \boldsymbol{0}_n, \end{cases} \qquad j = 1, 2, \ldots, n. \tag{15}$$

Then, by independence,

$$
\begin{aligned}
\mathbb{P}(\{X_j = 1\}) &= \mathbb{P}\left(\bigcap_{i=1}^{n}\left\{\boldsymbol{e}_i^\top\text{hardmax}(\boldsymbol{A})\boldsymbol{e}_j = 0\right\}\right) \\
&= \prod_{i=1}^{n} \mathbb{P}\left(\left\{\boldsymbol{e}_i^\top\text{hardmax}(\boldsymbol{A})\boldsymbol{e}_j = 0\right\}\right) \\
&= \prod_{i=1}^{n} (1 - p_{ij}). \tag{16}
\end{aligned}
$$

Therefore, the expectation number of columns with all zeros is

$$\mathbb{E}\left[\sum_{j=1}^{n} X_j\right] = \sum_{j=1}^{n} \mathbb{E}[X_j] = \sum_{j=1}^{n} \mathbb{P}(\{X_j = 1\}) = \sum_{j=1}^{n} \prod_{i=1}^{n} (1 - p_{ij}), \tag{17}$$

which completes the proof. $\square$

The required independence is provided by the following lemma.

**Lemma 4.** *[Vershynin (2018), Exercise 3.3.6] Let $\boldsymbol{G} \in \mathbb{R}^{m \times n}$ be a Gaussian random matrix, i.e. the entries of $\boldsymbol{G}$ are independent $\mathcal{N}(0, 1)$ random variables. Let $\boldsymbol{u}, \boldsymbol{v} \in \mathbb{R}^n$ be unit orthogonal vectors. Then, $\boldsymbol{G}\boldsymbol{u}$ and $\boldsymbol{G}\boldsymbol{v}$ are independent $\mathcal{N}(\boldsymbol{0}_m, \boldsymbol{I}_m)$ random vectors.*

*Proof.* First, we show that $\boldsymbol{G}\boldsymbol{u}, \boldsymbol{G}\boldsymbol{v}$ are both $\mathcal{N}(\boldsymbol{0}_m, \boldsymbol{I}_m)$ random vectors. This is straightforward since $\boldsymbol{G}\boldsymbol{e}_j \sim \mathcal{N}(\boldsymbol{0}_m, \boldsymbol{I}_m)$ gives $u_j \boldsymbol{G}\boldsymbol{e}_j \sim \mathcal{N}(\boldsymbol{0}_m, u_j^2 \boldsymbol{I}_m)$, and $\{u_j \boldsymbol{G}\boldsymbol{e}_j\}_{j=1}^n$ is a collection of independent Gaussian vectors. Hence $\boldsymbol{G}\boldsymbol{u} = \sum_{j=1}^n u_j \boldsymbol{G}\boldsymbol{e}_j \sim \mathcal{N}(\boldsymbol{0}_m, \|\boldsymbol{u}\|_2^2 \boldsymbol{I}_m)$.

Next, we show the independence of $\boldsymbol{G}\boldsymbol{u}$ and $\boldsymbol{G}\boldsymbol{v}$. Equivalently, we are supposed to prove that $\boldsymbol{e}_i^\top \boldsymbol{G}\boldsymbol{u}$ and $\boldsymbol{e}_{i'}^\top \boldsymbol{G}\boldsymbol{v}$ are independent random variables for any $i, i' \in [n]$. For $i \neq i'$, $(\boldsymbol{e}_i^\top \boldsymbol{G})\boldsymbol{u}$ and $(\boldsymbol{e}_{i'}^\top \boldsymbol{G})\boldsymbol{v}$ are independent random variables since $\boldsymbol{G}$ has independent rows. Therefore, the problem is reduced as the independence of $\boldsymbol{g}^\top \boldsymbol{u}$ and $\boldsymbol{g}^\top \boldsymbol{v}$ for $\boldsymbol{g} \sim \mathcal{N}(\boldsymbol{0}_n, \boldsymbol{I}_n)$. Notice that

$$[\boldsymbol{u}, \boldsymbol{v}]^\top \boldsymbol{g} \sim \mathcal{N}(\boldsymbol{0}_2, [\boldsymbol{u}, \boldsymbol{v}]^\top \boldsymbol{I}_n [\boldsymbol{u}, \boldsymbol{v}]) = \mathcal{N}(\boldsymbol{0}_2, \boldsymbol{I}_2), \tag{18}$$

which completes the proof. $\square$

Now we are ready to prove the main theorem.

*Proof of Theorem 1.* According to Lemma 4, since $\boldsymbol{x}_i^\top \boldsymbol{x}_j = \delta_{ij}$ (Kronecker symbol), $i, j = 1, 2, \cdots, n$, one can deduce that $\{\boldsymbol{q}_i\}_{i=1}^n = \{\boldsymbol{W}_q^\top \boldsymbol{x}_i\}_{i=1}^n$ is a collection of independent $\mathcal{N}(\boldsymbol{0}_{d_h}, \boldsymbol{I}_{d_h})$ random vectors. For any fixed Gaussian random matrix $\boldsymbol{W}_k$,

$$(\boldsymbol{e}_i^\top \boldsymbol{X}\boldsymbol{W}_q \boldsymbol{W}_k^\top \boldsymbol{X}^\top)^\top = \boldsymbol{K}\boldsymbol{q}_i \sim \mathcal{N}(\boldsymbol{0}_n, \boldsymbol{K}\boldsymbol{K}^\top), \tag{19}$$

which is also independent across different $i$'s. That is to say, the rows of $\boldsymbol{X}\boldsymbol{W}_q \boldsymbol{W}_k^\top \boldsymbol{X}^\top$ are independent and identically distributed as $\mathcal{N}(\boldsymbol{0}_n, \boldsymbol{K}\boldsymbol{K}^\top)$. Therefore, according to Lemma 3, the expectation number of columns with all zeros in $\mathrm{hardmax}(\boldsymbol{X}\boldsymbol{W}_q \boldsymbol{W}_k^\top \boldsymbol{X}^\top)$ is

$$\sum_{j=1}^n \prod_{i=1}^n (1 - p_{ij}) = \sum_{j=1}^n \prod_{i=1}^n (1 - p_j) = \sum_{j=1}^n (1 - p_j)^n. \tag{20}$$

Hence,

$$\frac{1}{n} \mathbb{E}_{\boldsymbol{W}_q} \left[ \mathrm{rank} \left( \mathrm{hardmax} \left( \boldsymbol{X}\boldsymbol{W}_q \boldsymbol{W}_k^\top \boldsymbol{X}^\top \right) \right) \right] \leq 1 - \frac{1}{n} \sum_{j=1}^n (1 - p_j)^n. \tag{21}$$

Let $p_j = c_j / n$, we get $\sum_{j=1}^n c_j = \sum_{j=1}^n n p_j = n \sum_{j=1}^n p_{ij} = n$. Let $\delta_j(n) := (1 - c_j/n)^n - \exp(-c_j)$. By Lemma 2 and Heine's theorem, we have $\lim_{n \to \infty} n \delta_j(n) = -c_j^2 \exp(-c_j)/2$. Notice that $g(y) := y^2 \exp(-y)$, $y \in (0, n)$ has a unique maximum $g(2) = 4 \exp(-2)$, we get

$$\frac{1}{n^2} \sum_{j=1}^n c_j^2 \exp(-c_j) \leq \frac{4 \exp(-2)}{n} < \frac{1}{n}. \tag{22}$$

Hence, for $\epsilon_0 > 0$ sufficiently small, there exists $n_0 \in \mathbb{N}_+$ such that for any $n \geq n_0$, we have

$$\left| \frac{1}{n} \sum_{j=1}^n \delta_j(n) \right| = \left| \frac{1}{n^2} \sum_{j=1}^n n \delta_j(n) \right| < \frac{1}{n}. \tag{23}$$

This gives

$$\frac{1}{n}\sum_{j=1}^{n}(1-p_j)^n = \frac{1}{n}\sum_{j=1}^{n}\exp\left(-c_j\right) + \frac{1}{n}\sum_{j=1}^{n}\delta_j(n)$$

$$\geq \left(\prod_{j=1}^{n}\exp\left(-c_j\right)\right)^{\frac{1}{n}} - \frac{1}{n}$$

$$= \left(\exp\left(-\sum_{j=1}^{n}c_j\right)\right)^{\frac{1}{n}} - \frac{1}{n}$$

$$= \exp\left(-1\right) - \frac{1}{n}, \tag{24}$$

where the AM-GM inequality is applied, and the equality holds if and only if $p_1 = p_2 = \cdots = p_n$. Hence, RHS of (21) $\leq 1 - \exp\left(-1\right) + 1/n$, which completes the proof. $\qquad\square$

