# OpenReview forum: "On the Efficiency of Transformers: The Effect of Attention Rank"
_ICLR.cc/2024/Conference — ICLR 2024 Conference Withdrawn Submission_

### Official Review · Reviewer_orQ3 · 2023-10-26

**Soundness:** 2 fair
**Presentation:** 3 good
**Contribution:** 2 fair
**Rating:** 5
**Confidence:** 4

**Summary:**

In this paper, the authors focus on studying the relationship between the mean rank of the attention matrices in Transformers and the model’s performance. The study is organised in the following steps:
* The authors show that the attention rank remains virtually unchanged during training. That is done by visualizing the statistics of interest during training GPT2- and BERT- like Transformer models;
* The authors demonstrate that having a higher rank attention correlates positively with a higher performance after training;
* It is shown that the attention temperature has considerably higher impact on rank than dimensionality of the head;
* The authors prove, under some assumptions, that there is upper bound on the initial attention rank;
* The above point is verified in a controlled setting.

**Strengths:**

* I think the paper proposes a novel angle of studying performance of Transformer models. There is a body of work looking at the attention ranks, yet to the best of my knowledge no papers were trying to tie those to final performance. I believe the mathematical result is novel.
* I believe the paper can be a source of an useful training/initialisation trick, if the authors can demonstrate a causal link between the rank at initialisation/temperature and the final performance.
* Modulo some caveats (see Weaknesses), I find the presentation to be sufficiently clear.

**Weaknesses:**

1. Upon reading the paper, my main question is whether the discovered relationship is causal, e.g. if starting with a higher initial attention rank/higher temperature leads to an improved performance. Indeed, if this holds, that paper has a high-impact recipe for improving the existing models. In the opposite case, I feel, an additional motivation for the study is required.
Importantly, here the ultimate experiment does not seem to be too hard to execute: one only needs to take some standard existing model with a standard training setup (e.g., BERT, which is not too large by modern standards) and, using insights from the paper, find a training regime that improves its final test loss.


2. S3.2: I am puzzled why only 4 train and 4 test examples were chosen for the illustrations. The plots represent averaged ranks, so why not having e.g. hundreds of points?
Furthermore, while we say that ranks do not change over training, the bottom curves almost universally go down as the training progresses.
Another puzzling observation is that all curves are somehow clustered in two groups: either at the very bottom or on the very top of the Figures 1 & 2/left-center. This could happen, for instance, if steps in the temperature are too high and the models either (a) work nicely, (b) get relatively broken. Would it be possible to have more fine-grained steps here?

3. Do the experimental results have another interpretation? What if the temperature has an impact on the performance directly, and that has nothing to do with the attention rank? Is there a way to refuse this interpretation?


Presentation:
1. As far as I can see, the details on how the rank is actually calculated only appear in S5.1 (SVD + threshold of 10^-8 on singular values). This should be mentioned earlier, before the first experiments that report rank measurements (S 3.2).
2. High-rank targets are mentioned in S1, yet are only defined at the end of the paper. It would be great to have a short explanation straightaway, otherwise I found the claim a bit puzzling on the first reading.
3. Almost everywhere the text uses \citet in place of \citep.
4. Capitalisation in ‘Transformers/transformers’ is not consistent throughout the paper (eg S1 vs S3 & S5).

**Questions:**

I would be happy to revisit my score if authors can add an experiment answering Question 1: whether we can use insights from the paper to improve training of some standard Transformer model. Alternatively, the authors can motivate the usefulness of their findings even if it is not causal.

---

### Official Review · Reviewer_cW94 · 2023-10-30

**Soundness:** 1 poor
**Presentation:** 2 fair
**Contribution:** 1 poor
**Rating:** 3
**Confidence:** 4

**Summary:**

This paper performs an analysis of the rank of attention during training and its effect in model validation loss on BERT and GPT-2 models. The main finding is that attention rank remains largely stable throughout training and that increasing the rank of the matrix leads to lower loss and faster convergence on IMDB and wiki datasets. In addition, it provides a theoretical analysis of the low-temperature case to come up with an upper bound for the expected rank by making specific assumptions (independent k,q and hardmax activation). Experiments on a toy task show that increasing the value of T leads to better performance.

**Strengths:**

- Investigating the rank of weight matrices from large language models can provide insights on their expressivity, compressibility, and the type of representations the model is learning during the training process.
- The finding that the rank remains stable throughout the training procedure could inform ways to tune the softmax temperature efficiently to reach a desired rank value and avoid poor starting points.

**Weaknesses:**

- Motivation for the empirical/theoretical analysis and implications of the associate findings is not very clear; the key contributions and proper positioning with respect to prior work is also lacking. A few of the observations made in the theoretical analysis and numerical validation are well-known for softmax with temperature hyper-parameter e.g. that it can sharpen and flatten the distribution. The low temperature setting leads to concentrated attention i.e. low-rank and in the high temperature leads to high-rank (i.e. uniform weights).
- In terms of novelty, the analysis of rank in weight matrices has been explored in the compression-related literature for transformers e.g. [1, 2], It is not very well defined what is new here and how it relates to efficiency of transformers; the paper doesn't study efficiency-related  properties of transformers as the title suggests.
- The main finding of the paper that increasing the rank through temperature scaling leads to higher performance is counter-intuitive. When we increase the rank, in the limit the softmax weights become uniform i.e. the role of learned attention weights gets ignored.
- The claim that the increase in rank leads to better performance is not well supported; results are based on validation loss reported on two datasets and a toy task. It's unclear if the results would generalize to other tasks and different model scales.
- The assumptions made in the theoretical part are not necessarily valid for the real scenarios of transformers with softmax (instead of hardmax) and learned queries & keys (they are not necessarily independent).

[1] https://arxiv.org/pdf/2207.00112.pdf
[2] https://openreview.net/pdf?id=_sSHg203jSu

**Questions:**

- Can the authors explain what is the usefulness of having a rank that is higher than the dimension d in Fig. 5? The softmax bottleneck dictates that the inner dimension of the factorization provides an upper-bound for the rank. By increasing the temperature to a very high-value we effectively create a uniform distribution that will always have rank equal to the sequence length but this is equivalent to ignoring the learned distribution by the model. It's unclear to me what is the benefit of this.
- What is the definition of "efficiency" and "marginal utility" after Eq. 8?  It was also not clear to me how the initial part from the this equation was derived; can the authors explain?
- What is the task and training objective for IMDB and Wiki datasets? In the IMDB case, it would be useful to report the accuracy instead of the validation loss. I'd recommend repeat the experiments on a larger number of tasks to showcase that the results are broadly applicable.

---

### Official Review · Reviewer_ZXWy · 2023-11-01

**Soundness:** 2 fair
**Presentation:** 2 fair
**Contribution:** 2 fair
**Rating:** 3
**Confidence:** 4

**Summary:**

This paper studies the rank of attention matrices during training, how the attention temperature and attention head dimensions impact the rank, and how the rank correlates with model performance. It shows that the attention rank remains stable during training, that higher temperature generally leads to higher rank, and that head dimension has a more minor impact on rank. Experiments were performed with BERT and GPT-2 models, training on Wiki and IMDB datasets. Theoretically, the papers proves that the expected rank of a (hardmax) attention matrix with orthonormal input data, and random $W_q$ and $W_k$ matrices is $\leq (1- 1/e)n +1 \approx 0.63n$, where $n$ is the number of rows in the data matrix $X$.

**Strengths:**

- I found the theorem related to the rank of a hardmax attention matrix with random $W_k$ and $W_q$ matrices mathematically interesting.
- I found it interesting that the rank of the attention matrices remained quite stable during training.

**Weaknesses:**

- The paper implies that training with temperature $T=\sqrt{d_h}$ yields better performance than training with temperature $T=0.001 * \sqrt{d_h}$ **because** the attention matrix is higher rank with the higher temperature. However, I am unconvinced that this relationship is **causal**. Perhaps training with the higher temperature yields better performance due to optimization issues, for example. If the relationship were causal, it seems we should expect to see better validation loss for higher head dimensions (with low temp), which is not the case.
- The paper does not demonstrate that its insights related to attention rank can be used to make any meaningful improvements to transformer model architectures or training algorithms. The experiments in section 5.2 are very small-scale and overly synthetic, and thus unconvincing.
- It doesn’t make sense to me that for very large temperature values, the attention matrix approaches full-rank (Figure 5). In particular, for $T=\infty$, the logits would all be 0, and thus all entries in the attention matrix would be equal to $1/\sqrt{n}$, which is a rank 1 matrix.
- The definition of rank that is used in the experiments (# singular values > $10^{−8}$) should be made clear much earlier (currently defined on page 7).
- It is unclear to me what the practical consequences of Theorem 1 are.

**Questions:**

- How was the threshold of $10^{-8}$ chosen? Why is this a good definition of rank? Did you consider other ways of measuring rank?